# GROUP-DISENTANGLING CONDITIONAL SHIFT

## ABSTRACT

We propose a novel group disentanglement method called the Context-Aware Variational Autoencoder (CxVAE). Our model can learn disentangled representations on datasets with conditional shift. This phenomenon occurs when the distribution of the instance-level latent variable $\mathbf{z}$ conditional on the input observation $\mathbf{x}$, $p(\mathbf{z}|\mathbf{x})$, changes from one group to another (i.e. $p_i(\mathbf{z}|\mathbf{x}) \neq p_j(\mathbf{z}|\mathbf{x})$, where $i,j$ are two different groups). We show that existing methods fail to learn disentangled representations under this scenario because they infer the group $\mathbf{u}$ and instance $\mathbf{z}$ representations separately. CxVAE overcomes this limitation by conditioning the instance inference on the group variable $q(\mathbf{z}|\mathbf{x},\mathbf{u})$. Our model has the novel ability to disentangle ambiguous observations (those with incomplete information about the generative factors), which we evaluate on an image dataset. Additionally, we use a fair comparisons task to demonstrate empirically that conditional shift is the cause of our model's improved performance.

## 1 INTRODUCTION

Group disentanglement is the goal of learning representations that separate group-level variation from instance-level variation. Consider a dataset of observations organised into $N$ groups of the form $\mathbf{x}_{n,1:K_n} = \{\mathbf{x}_{n,1}, ..., \mathbf{x}_{n,K_n}\}$, $n \in 1 : N$. These could be pictures grouped by author, clinical outcomes grouped by the patient, or film ratings grouped by user. We train a representation network $r(\mathbf{x}_n)$ that encodes a group of observations $\{\mathbf{x}_{n,1}, \ldots \mathbf{x}_{n,K_n}\}$ into one group code $\mathbf{u}_n$ and a set of instance codes $\{\mathbf{z}_{n,1}, \ldots \mathbf{z}_{n,K_n}\}$, one for each observation. We want $\mathbf{u}$ to capture only the variation across groups and $\mathbf{z}$ only the variation within groups.

The current state-of-the-art approaches for group disentanglement train the representation network $r$ by using it as the variational latent posterior distribution in a Variational Autoencoder (Bouchacourt et al., 2018; Hosoya, 2019; Németh, 2020). They assume a hierarchical generative model whereby the observation $\mathbf{x}_{n,k}$ is generated by combining a group latent variable $\mathbf{u}_n$ and an independent instance latent variable $\mathbf{z}_{n,k}$ (Figure 2-left). The standard setup involves training the variational latent posterior $q(\mathbf{u}_n, \mathbf{z}_{n,1:K_n}|\mathbf{x}_n)$ by maximising a lower bound to the data likelihood (Kingma & Welling, 2014; Rezende et al., 2014).

In our work, we show that the variational latent posterior, as defined in existing models, is unsuited to datasets with *conditional shift*. This is a property of the data-generating process whereby the true conditional distribution of the instance latent variable $\mathbf{z}$ changes from one group to another $p_i(\mathbf{z}|\mathbf{x}) \neq p_j(\mathbf{z}|\mathbf{x})$ where $i,j$ are two groups (Zhang et al., 2013; Gong et al., 2016). In our case, the conditional instance distribution for group $i$, which is $p_i(\mathbf{z}|\mathbf{x})$, corresponds to $p(\mathbf{z}_{i,k}|\mathbf{x}_{i,k}, \mathbf{u}_i)$ where $k$ is a given instance in the group.

Conditional shift occurs in many real-world datasets that we would like to group-disentangle. For example, in the 3DIdent dataset (Zimmermann et al., 2021), if we want to infer the colour of the teapot $\mathbf{z}_{n,k}$ based on an image of that teapot $\mathbf{x}_{n,k}$, we should take into account the colour of the spotlight that illuminates the scene $\mathbf{u}_n$; different coloured spotlights will make the same object appear different colours, as can be seen in Figure 1. Existing group-disentanglement methods, which infer the instance variable (teapot colour) independently of the group (spotlight colour) fail to disentangle the two colours.

Existing VAE-based methods fail to disentangle in the conditional shift setting because they make the assumption that the group and instance variables can be inferred independently of each other from the input observation (Figure 2-middle): When defining the variational latent posterior, existing works

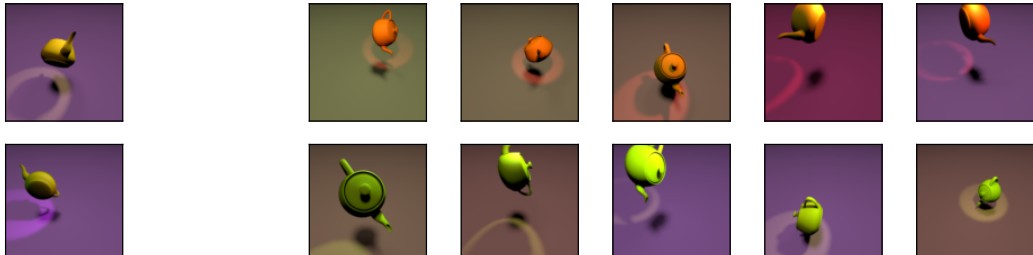

Figure 1: **Conditional shift in the 3DIdent dataset (Zimmermann et al., 2021).** The two images on the leftmost column appear to show objects of the same colour, even though each image was generated by a different combination of object-colour, spotlight-colour: One teapot is actually orange, while the other is bright green. We can see this by looking at other examples of the same objects under different lighting. The set of images to the right depict different views of the same object on each corresponding row.

based on the Group VAE (GVAE) (Bouchacourt et al., 2018; Hosoya, 2019; Németh, 2020; Chen & Batmanghelich, 2020) make the assumption that the group and instance variables are conditionally independent given the observations (Figure 2-middle):

$$q(\mathbf{u}_n, \mathbf{z}_{n,1:K_n} | \mathbf{x}_{n,1:K_n}) = q(\mathbf{u}_n | \mathbf{x}_{n,1:K_n}) \sum_{k=1}^{K_n} q(\mathbf{z}_{n,k} | \mathbf{x}_{n,k}). \tag{1}$$

The limitations of this assumption have not been identified so far in the literature because the datasets used to test disentanglement, such as Shapes3D (Kim & Mnih, 2018), SmallNORB (LeCun et al., 2004), dSprites (Higgins et al., 2017), Cars3D (Reed et al., 2015), MPI3D (Gondal et al., 2019), have the property that one image is always sufficient to accurately infer its latent variables. For example, we only require a single image from the MPI3D dataset to uniquely identify the colour, position, and rotation of the depicted object.

In this work, we show that conditioning the instance encoder on the group latent vector enables the model to learn disentangled representations on datasets with conditional shift.

1. In the first instance, we show that only our method is able to correctly disentangle between object-colour and spotlight-colour in the 3DIdent dataset (Zimmermann et al., 2021), illustrated in Figure 1.

2. Then, we use the task of fair comparisons between student test-scores (Figure 3) to show that the amount of conditional shift in the dataset determines the performance gap between our model and the other approaches.

## 2 RELATED WORK

**Group Disentanglement.** This class of problems comes under different names: style-content disentanglement (Tenenbaum & Freeman, 2000), content-transformation disentanglement (Hosoya, 2019), and disentanglement with group supervision (Shu et al., 2020), to name a few. Recent work (Shu et al., 2020; Locatello et al., 2020) has contextualised group disentanglement as a subproblem of weakly-supervised disentanglement, where disentangled representations are learned with the help of non-datapoint supervision (e.g. grouping, ranking, restricted labelling). Early work in this area focused on separating between visual concepts (Kulkarni et al., 2015; Reed et al., 2015). This area has received renewed interest after the theoretical impossibility result of Locatello et al. (2019) and the identifiability proofs of Khemakhem et al. (2020) and Mita et al. (2021). A key aspect of recent weakly-supervised models is the interpretation of the grouping as a signal of similarity between datapoints (Chen & Batmanghelich, 2020).

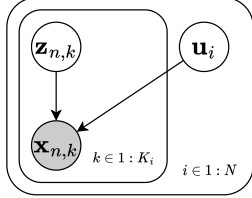 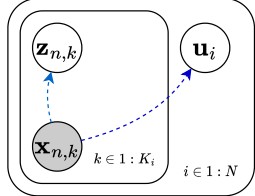 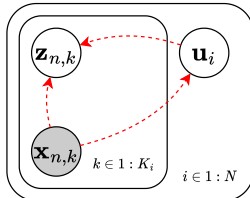

(a) Group generative model    (b) Inference model of existing method GVAE (Hosoya, 2019)    (c) Our inference model (CxVAE)

Figure 2: Our variational distribution conditions the instance representation $\mathbf{z}_{n,k}$ on the group representation $\mathbf{u}_n$.

**Conditioning on the Group Variable.**    While conditioning the instance encoder on the group variable is common in the areas of semi-supervised learning and fair representations (Kingma et al., 2014; Louizos et al., 2016), we are the first to apply it to unsupervised group disentanglement where explicit group labels are not available. In the field of sequence disentanglement, state-of-the-art methods (Hsu et al., 2017; Denton & Birodkar, 2017; Li & Mandt, 2018) infer the instance variable (capturing shorter timescales) conditionally on the group variable (capturing longer timescales). Recent works in weakly-supervised disentanglement (Shu et al., 2020; Locatello et al., 2019; Roeder et al., 2019) also condition the instance variable on the group, but their group variable is a discrete variable used for selection rather than a representation. It marks which units of the instance representation are common within the group and which are free to vary. We argue that this is not sufficient to account for the variation in $p(v|x, u)$ produced by the conditional shift, so we include AdaGVAE (Locatello et al., 2020) in our evaluation for comparison (see Section 7).

**Conditional Shift.**    Our instance encoder conditioned on the group variable is a new strategy to deal with conditional shift. This problem has been studied extensively in the context of supervised learning (Zhang et al., 2013; Gong et al., 2016). However, we are the first to explore the effect of conditional shift on unsupervised learning. Methods for mitigating the effects of conditional shift typically focus on learning domain-invariant representations (Ben-David et al., 2009). However, Zhao et al. (2019) show that learning a domain-invariant representation is not sufficient for learning a correct mapping between instance variables from different groups.

**Image Translation.**    Note that our variational latent posterior is different from the one used in COCO-FUNIT (Saito et al., 2020). The authors are motivated by the same limitations with existing works as we are, namely that unsupervised translation methods struggle to disentangle under conditional shift. However, because they train explicitly for translation rather than disentanglement, they arrive at a different solution than ours. When performing a translation, their approach is to condition the representation of the target group on the source image, thereby bypassing the need for an accurate instance representation. This mechanism produces impressive results on image translation tasks, but it cannot be extended to models based on the GVAE which do not train explicitly for translation; in our case, there are no source and target groups in the training set. Regardless, we evaluate COCO-FUNIT on the test score dataset and show that our model outperforms it both in terms of disentanglement and translation.

## 3 BACKGROUND

The Group-Instance Generative Model (Bouchacourt et al., 2018; Hosoya, 2019) is a multi-level model that uses two latent variables to generate grouped data: the instance variable $\mathbf{z}_{n,k} \sim \mathcal{N}(0, 1)$ controls the variation within groups, and the group variable $\mathbf{u}_n \sim \mathcal{N}(0, 1)$ controls the variation across groups (Figure 2-left). The likelihood of a group $\mathbf{x}_{n,1:K_n}$ is:

$$p(\mathbf{x}_{n,1:K_n}) = \mathbb{E}_{p(\mathbf{u}_n)} \prod_{k=1}^{K_n} \mathbb{E}_{p(\mathbf{z}_{n,k})} \left[ p(\mathbf{x}_{n,k}|\mathbf{u}_n, \mathbf{z}_{n,k}) \right]$$

### 3.1 VARIATIONAL INFERENCE

Because the exact likelihood is intractable, the standard approach to train the group-instance generative model is with a Variational Autoencoder (Kingma & Welling, 2014; Rezende et al., 2014) which performs optimisation by introducing a variational latent posterior $q(\mathbf{u}_n, \mathbf{z}_{n,1:K_n}|\mathbf{x})$ and maximizing the Evidence Lower Bound (Jordan et al., 2004):

$$
\begin{aligned}
\log p(\mathbf{x}_{n,1:K_n}) \geq \; & \mathbb{E}_{q(\mathbf{u}_n, \mathbf{z}_{n,1:K_n}|\mathbf{x}_{n,1:K_n})} \left[ \sum_{k=1}^{K_n} \log p(\mathbf{x}|\mathbf{u}, \mathbf{z}) \right] \\
& - \mathrm{KL}[q(\mathbf{u}_n, \mathbf{z}_{n,1:K_n}|\mathbf{x}_{n,1:K_n})||p(\mathbf{u}_n, \mathbf{z}_{n,1:K_n})]
\end{aligned}
\tag{2}
$$

Existing methods use a class of variational distributions that assume conditional independence between the latent variables.

$$
q(\mathbf{u}_n, \mathbf{z}_{n,1:K_n}|\mathbf{x}_{n,1:K_n}) = q(\mathbf{u}_n|\mathbf{x}_{n,1:K_n}) \prod_{k=1}^{K_n} q(\mathbf{z}_{n,k}|\mathbf{x}_{n,k})
$$

## 4 CONTEXT-AWARE VARIATIONAL AUTOENCODER

We propose a new model which can perform well on group-confounded problems. We call our model the Context-Aware Variational Autoencoder (CxVAE). Its defining feature is a variational latent posterior whose instance variable is conditioned on the group variable:

$$
q(\mathbf{u}_n, \mathbf{z}_{n,1:K_n}|\mathbf{x}_{n,1:K_n}) = q(\mathbf{u}_n|\mathbf{x}_{n,1:K_n}) \prod_{k=1}^{K_n} q(\mathbf{z}_{n,k}|\mathbf{x}_{n,k}, \mathbf{u}_n)
$$

Thus, the instance encoder is implemented as a network which takes as input the concatenation of the observation $\mathbf{x}_{n,k}$ and previously sampled group representation $\mathbf{u}_n$.

$$
q(\mathbf{z}_{n,k}|\mathbf{x}_{n,k}, \mathbf{u}_n) = \mathcal{N}(\mu, \sigma), \quad (\mu, \sigma) = f(\mathbf{x}_{n,k}, \mathbf{u}_n),
$$

This form of the variational distribution reflects the correct factorisation of the generative model. It has the potential to learn the true generative latent posterior, which is disentangled by definition. The Evidence Lower Bound for our model is

$$
\begin{aligned}
\mathrm{ELBO}(\mathbf{x}_{n,1:K_n}) = \; & \mathbb{E}_{q(\mathbf{u}_n, \mathbf{z}_{n,1:K_n}|)} \left[ \sum_{k=1}^{K_n} \log p(\mathbf{x}_{n,k}|\mathbf{u}_n, \mathbf{z}_{n,k}) \right] \\
& - \mathrm{KL}[q(\mathbf{u}_n|\mathbf{x}_{n,1:K_n})||p(\mathbf{u}_n)] \\
& - \mathbb{E}_{q(\mathbf{u}_n|\mathbf{x}_{n,1:K_n})} \left[ \sum_{k=1}^{K_n} \mathrm{KL}[q(\mathbf{z}_{n,k}|\mathbf{x}_{n,k}, \mathbf{u}_n)||p(\mathbf{z}_{n,k})] \right].
\end{aligned}
\tag{3}
$$

## 5 EVALUATION METHODOLOGY

We show that by making the instance encoder conditional on the inferred group variable, we obtain a considerable gain in translation accuracy and a marked improvement in disentanglement. We also demonstrate that the gap in performance between our model and other group disentanglement methods is caused by conditional shift in the data-generating process.

## 5.1 MODEL SETUP

We compare our conditional CxVAE with the state-of-the-art in group disentanglement, namely the Group VAE (Hosoya, 2019; Bouchacourt et al., 2018), COCO-FUNIT (Saito et al., 2020), and AdaGVAE (Locatello et al., 2019). As in Hosoya (2019), the group encoder is applied to each datapoint in the group and then all the outputs are averaged.

For all experiments, our CxVAE will be a modified GVAE such that the group variable $\mathbf{u}_n$ is concatenated with the observation $\mathbf{x}_{n,k}$ and fed into the instance encoder in order to compute the instance variable $\mathbf{z}_{n,k}$. For sampling the variational latent posteriors, we use the standard reparametrisation trick. We use an Adam optimiser with learning rate of 1e-4 with $\beta_1 = 0.9, \beta_2 = 0.5$.

For the 3DIdent dataset, we implement all networks (encoders and decoders) as convolutional nets with 4 hidden layers and 64 filters each. Both latent variables have 16 latent dimensions.

For the test-score dataset, we use MLPs with 3 hidden layer of 32 activations each. The group variable will have 4 dimensions and the instance variable will have 2 dimension.

We train each model for 64 epochs, and use the last 10 epochs for evaluation. Additionally, we run the experiment for 100 different random seeds initialisations, both for the data generating process and the networks. Confidence intervals are computed by resampling train-test splits, weight initialisations and sampling seeds. We use the same 100 seeds in each model. This gives 1000 measurements to plot in Table 1.

## 5.2 EVALUATION METRICS

We compare the models with respect to 3 different criteria: 1) How well does the model fit the holdout data?, 2) How disentangled are the representations inferred by the encoder?, and 3) How well can the model answer the counterfactual question "What colour would appear on the screen if this object were lit by a different spotlight?". We assess each criterion with a different evaluation metric.

**Fitting the holdout data.** As a general metric, we report the reconstruction error (MSE) on the holdout data for every experiment, commonly used as a proxy for the likelihood of the holdout set.

**Disentangled representations.** We use the Mutual Information Gap (Chen et al., 2018) to measure the quality of the disentanglement. We measure empirically the amount of mutual information between the inferred latent variables $\mathbf{u}, \mathbf{z}$ and the ground-truth group-level factor $\mathbf{u}'$. Consequently, the goal is to have maximal mutual information between the group variable $\mathbf{u}$ and the ground-truth $\mathbf{u}'$, and minimal mutual information between the instance variables $\mathbf{z}$ and the ground-truth $\mathbf{u}'$. The gap between the two (normalized with the entropy of the ground-truth factors) is the metric of disentanglement:

$$\text{MIG} = \frac{1}{H(\mathbf{u}')}(I(\mathbf{u}; \mathbf{u}') - I(\mathbf{z}; \mathbf{u}')) \tag{4}$$

Since the data-generating process is known, the mutual information between the inferred group variable and the ground-truth group variable $I(\mathbf{u}; \mathbf{u}')$ is straightforward to implement by following the approach from Chen et al. (2018).

We measure only the mutual information between the ground-truth group factor $\mathbf{u}'$ and the latent variables because, as pointed out by Németh (2020), the common failure case we are trying to guard against in group disentanglement is that the instance variables $\mathbf{z}$ might learn information belonging to the ground-truth group factor $\mathbf{u}'$.

**Translation task.** We measure how well the learned representations can answer the question "What would the score of student $k$ from school $n$ have been if they had attended the typical school (the school with scores distributed according to $\mathcal{N}(0, I_2)$)?". This problem, also known as translation, is a commonly used downstream task for disentangled representations (Tenenbaum & Freeman, 2000). We translate the score of student $k$ to the typical school, and then take the mean squared error against the ground-truth translation, which is generated when the data is generated using the ground-truth

Table 1: Our CxVAE outperforms existing methods at disentangling between object-colour and spotlight-colour in the 3DIdent (Zimmermann et al., 2021) and our test-score datasets. Our model's MIG scores is much higher than the one produced by competing models, and the performance gap is greater than the 95% confidence interval for any one of these models. Confidence intervals are computed by resampling train-test splits, weight initialisations and sampling seeds.

| MODEL | RECONSTRUCTION | MIG | TRANSLATION |
|---|---|---|---|
| **3DIdent (Zimmermann et al., 2021)** | | | |
| GVAE (Hosoya, 2019) | $0.08 \pm 0.02$ | $0.09 \pm 0.06$ | $0.23 \pm 0.18$ |
| AdaGVAE (Locatello et al., 2020) | $0.09 \pm 0.05$ | $0.39 \pm 0.11$ | $0.15 \pm 0.05$ |
| COCO-FUNIT (Saito et al., 2020) | $\mathbf{0.05} \pm 0.01$ | $0.13 \pm 0.04$ | $0.13 \pm 0.06$ |
| **CxVAE (ours)** | $\mathbf{0.05} \pm 0.03$ | $\mathbf{0.72} \pm 0.06$ | $\mathbf{0.08} \pm 0.05$ |
| **Student test scores** | | | |
| GVAE (Hosoya, 2019) | $0.41 \pm 0.01$ | $0.04 \pm 0.02$ | $0.49 \pm 0.01$ |
| AdaGVAE (Locatello et al., 2020) | $0.41 \pm 0.01$ | $0.05 \pm 0.02$ | $0.49 \pm 0.01$ |
| COCO-FUNIT (Saito et al., 2020) | $0.40 \pm 0.02$ | $0.04 \pm 0.01$ | $0.49 \pm 0.02$ |
| **CxVAE (ours)** | $\mathbf{0.35} \pm 0.02$ | $\mathbf{0.44} \pm 0.08$ | $\mathbf{0.37} \pm 0.02$ |
| **lower is better** | **higher is better** | **lower is better** | |

generative factors. For our dataset, the correct translation corresponds to the Earth-Mover distance between the multivariate normal distributions of scores in each school (Knott & Smith, 1984).

In order to obtain the translation, we first infer the instance variable of student $k$ from school $n$ and the group variable of the typical school. We then feed the two variables to the decoder. For evaluation, we translate all the scores from each school $n$ and then measure the distance between the predicted translation and the ground-truth translation. We compute the total error as an average over all the translation errors.

We can also use translation as an additional qualitative comparison between CxVAE and other group disentanglement methods, which can be seen in Figure 3.

## 6 DISENTANGLING AMBIGUOUS OBSERVATIONS

We show that our CxVAE is able to group-disentangle in a setting where conditional shift produces ambiguous observations. The dataset comprises images from the 3DIdent dataset (Zimmermann et al., 2021) depicting teapots of different colours being lit by spotlights of different colours. Within a group, the images have the same spotlight colour and only the colour of the teapot varies. The goal is for the group representation to encode the spotlight colour and for the instance representation to encode the object colour.

The goal of disentangling between object-colour and spotlight-colour is useful for many real-world applications, such as object recognition. Once we have separated the group-level variation from the instance level variation, we can use the instance representation as a low-level feature on which to train a classifier that predicts the colour of the object. The better the disentanglement, the better performance this predictor would have at identifying the true colour of the object.

This is a difficult problem because the data exhibits conditional shift. The same exact image could have been generated by different combination of spotlight colour and object colour, as can be seen in Figure 1. This makes it difficult to identify the true colour of the object by just looking at one single image.Indeed all previous methods, which infer the instance representation solely from the current image, fail to learn disentangled representations of this dataset.

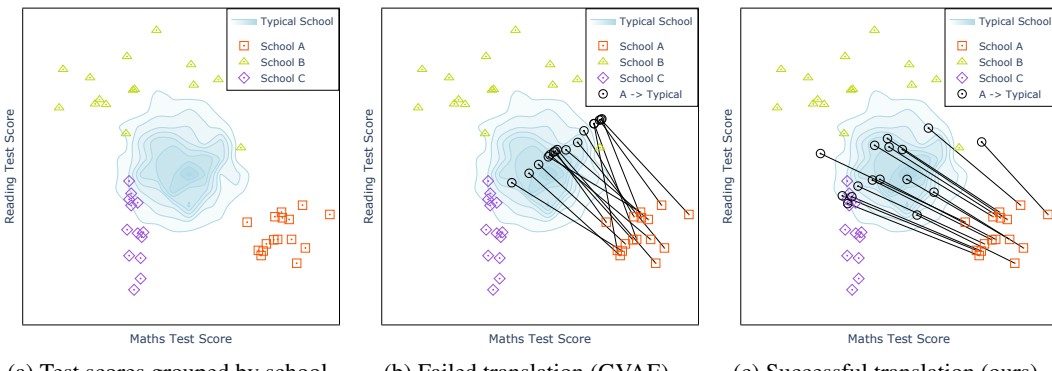

(a) Test scores grouped by school.  (b) Failed translation (GVAE).  (c) Successful translation (ours).

Figure 3: **Our model correctly disentangles between student aptitude and school characteristics, whereas the GVAE (Hosoya, 2019) fails.** We ask the counterfactual question "What score would student $k$ from school $A$ have obtained if they had attended the *typical school*?". The task is to generate a set of test scores by translating the scores from school $A$ onto the distribution of scores from the typical school. Each line shows the translation for an individual student. Our model preserves the relative positions of the scores, whilst also capturing the distribution of the typical school.

The results in Table 1 show that our model, the CxVAE, produces representations that disentangle between spotlight colour and object colour much more than a representative selection of existing models: GVAE, AdaGVAE, and COCO-FUNIT. Our model's MIG score is much higher than the one produced by competing models, and the performance gap is greater than the 95% confidence interval for any one of these models.

## 7 CONDITIONAL SHIFT CAUSES THE GAP IN PERFORMANCE

Our CxVAE produces considerable improvements over the competing methods in terms of fitting the holdout set, disentangled representations, translation accuracy and predicting the generative factors (Table 1). While the scores of the existing methods cluster together, the gap between them and the CxVAE is larger than the 95% confidence interval of any method.

### 7.1 FAIR COMPARISONS BETWEEN STUDENTS

Consider the task of fair comparisons between students attending different schools based on their standardised test scores in maths and reading (Braun et al., 2006). The typical assumption in the literature is that each school has a similar distribution of aptitude among its students, so our goal is to learn disentangled student-level (instance) and school-level (group) representations of the scores. The instance representation $\mathbf{z}_{n,k}$ should reflect the aptitude of student $k$ from school $n$ independently of what school the student had attended.

This analysis is crucial for university admission boards aiming to judge students based on their aptitude regardless of the socio-economic circumstances associated with attending one school or another, such as affluence, location, or curriculum (Raudenbush & Willms, 1995; Braun et al., 2006). Group disentanglement has the potential to reduce the computational costs of this analysis compared to the current variational inference methods for mixed-effects models (Gelman & Hill, 2006). The variational latent posterior of a GVAE can perform scalable inference of the student and school representations requiring just one forward pass through the network for every additional student. Contrast this with running a separate optimisation routine for the latent variables of each student (Gelman et al., 1995). GVAEs can also optimise highly non-linear generative models which would otherwise have to be designed explicitly for the problem in hand (Pinheiro & Bates, 2001).

Current methods fail to learn disentangled representations on this data. Figure 3b shows the GVAE model (Hosoya, 2019) incorrectly translating the scores from one school to another. Translation is a well-established downstream task for evaluating disentanglement (Tenenbaum & Freeman, 2000).

In the case of test scores, translation corresponds to the counterfactual question "What score would student $k$ from school $A$ have obtained if they had attended the *typical school* (i.e. a school whose scores are distributed according to $\mathcal{N}(0, I_2)$)?". We do this by generating a new score by combining the instance representation of student $k$ from school $n$ with the group representation of the typical school. Raudenbush & Willms (1995) use translation to directly compare students from different schools. Notice that the unconditional GVAE scrambles the order of the scores and fails to capture the $\mathcal{N}(0, I_2)$ distribution of the typical school.

We propose a modification to the variational latent posterior in order to enable the GVAE to learn disentangled representations of test score data. Looking at the data (Figure 3a), it is clear that we are dealing with the conditional shift scenario. The same reading score could be obtained by either a high-achieving student from school $C$ or a low-achieving student from school $B$. Inferring the aptitude of the student requires knowing the distribution of scores within each school. In order to account for the variation across groups, we introduce the Context-Aware Variational Autoencoder (CxVAE), whose instance encoder is conditioned on the group representation (Figure 2-right). This reflects the correct factorisation of the generative latent posterior $p(\mathbf{u}_n, \mathbf{z}_{n,1:K_n} | \mathbf{x}_{n,1:K_n})$, thus making no assumptions about the relationship between the group and the instance. We observe our variational model successfully translating test scores in Figure 3c.

## 7.2 DATASET

We generate our dataset of test scores using the classic "varying intercept, varying slope" mixed-effects model (Laird & Ware, 1982; Pinheiro & Bates, 2001; Gelman & Hill, 2006). This is a well-established approach for modelling student scores $\mathbf{x}_{n,k}$ as a function of individual aptitude $\mathbf{a}_{n,k}$ and school-level characteristics ($\mathbf{b}_n, \mathbf{c}_n$) (e.g. affluence, curriculum, location, etc.) (Raudenbush & Willms, 1995; Braun et al., 2006). We choose this model for its simplicity and for the wide variety of phenomena to which it can be applied. All the scores and factors are 2-dimensional vectors, with one component for the maths score and another for the reading score:

$$
\begin{aligned}
&\mathbf{x}_{n,k} = \mathbf{b}_n - \mathbf{c}_n \odot \mathbf{a}_{n,k} + \epsilon_{n,k} \ \text{ is the score of student } k \text{ in school } n. \\
&\mathbf{a}_{n,k} \sim \mathcal{N}(0, I_2) \ \text{ is the aptitude of student } k \text{ in school } n \\
&\mathbf{b}_n \sim \mathcal{N}(0, I_2) \ \text{ is the mean score in school } n \\
&\mathbf{c}_n \sim \text{Exp}(1) \ \text{ is the standard deviation of scores in school } n \\
&\epsilon_{n,k} \sim \mathcal{N}(0, 0.1 * I_2) \ \text{ is a per-student error term.}
\end{aligned}
\tag{5}
$$

For the evaluation procedure, we use the above model to generate $N = 32,768$ values for $\mathbf{b}_n, \mathbf{c}_n$. For each school $n$, we generate $M = 128$ values for $\mathbf{a}_{n,k}$. We then randomly select half of the schools to assemble a training dataset with $2,097,152$ scores split across $16,384$ schools. We take the other half of schools to create the holdout dataset, so that every testing school and student are unseen during training.

We have chosen to evaluate on a synthetic dataset because it allows for fine control over the parameters of the data-generating process (especially relevant in Section 7.3) and it also enables us to measure the quality of disentanglement using the Mutual Information Gap (Chen et al., 2018).

## 7.3 CONDITIONAL SHIFT CAUSES THE GAP IN PERFORMANCE

We show, by modifying the data-generating process (5), that conditional shift explains the increased performance of the CxVAE. We insert a hyper-parameter $\lambda$ to control the strength of the conditional shift; $\lambda = 1$ means the conditional shift stays the same as in the previous experiment, while $\lambda = 0$ means there is no conditional shift.

Consider the case where the maths score only depends on the school, and the reading score only depends on the student. In this situation, the two generative factors can be easily disentangled since you can infer the student aptitude from the reading score and the school profile from the maths score. We use this as an extreme case of lack of conditional shift, and insert a hyper-parameter $\lambda$ in our data-generating process that will continuously move between this case and the original case. Our modified data-generating process is

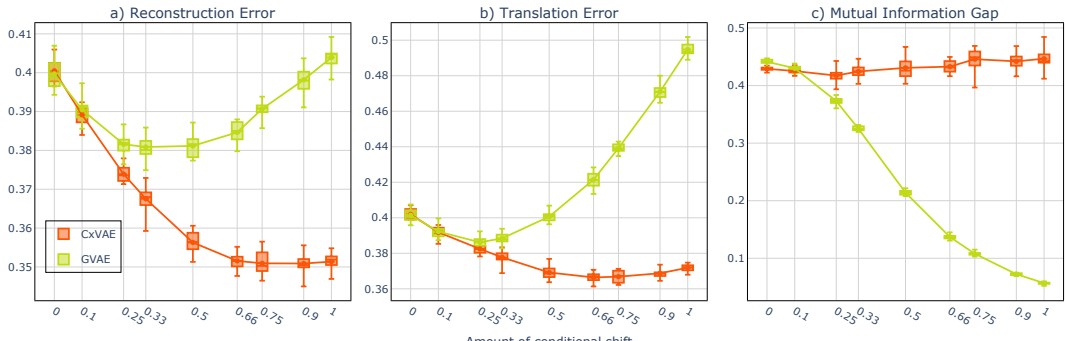

Figure 4: **Conditional shift in the dataset causes the relative performance gain of our CxVAE over the GVAE.** We show performance on datasets generated with different values of the $\lambda$ hyperparameter, controlling the amount of conditional shift. For low values of $\lambda$ the conditional distribution of student aptitude given a score changed very little from one group to another, and so the CxVAE and GVAE perform equally well. However, as $\lambda$ increases, the GVAE fails to disentangle.

$$\mathbf{x}_{n,k} = \begin{bmatrix} \lambda \\ 1 \end{bmatrix} \odot \mathbf{b}_n + \begin{bmatrix} 1 \\ \lambda \end{bmatrix} \odot \mathbf{a}_{n,k} \odot \left( c_n \hat{} \begin{bmatrix} \lambda \\ 1 \end{bmatrix} \right) + \epsilon_{n,k} \tag{6}$$

where $\hat{}$ denotes the elementwise power operation and the generative factors $(\mathbf{a}_{n,k}, \mathbf{b}_n, \mathbf{c}_n, \epsilon_{n,k})$ are sampled as in (5).

This model has no conditional shift when $\lambda = 0$ because each ground-truth factor controls a separate component of the data. Inferring the student aptitude requires only the reading score and can ignore the school characteristics. When $\lambda = 1$, the problem exhibits conditional shift in exactly the same way as in (5).

If our hypothesis is correct, that the conditional shift causes the performance gap between CxVAE and other group disentanglement methods, then the gap should decrease as $\lambda$ approaches 0. The measurements displayed in Figure 4 confirm our expectations. For low values of $\lambda$ the performance of our CxVAE is evenly matched to the GVAE. As $\lambda$ increases, CxVAE metrics remain stable while GVAE performance decreases substantially. It is clear that the degree of confounding in the dataset explains the performance gain that we see in the CxVAE.

## 8 CONCLUSIONS

In this work, we show empirically that conditioning the instance encoder on the group variable produces group-disentangled representations on datasets with conditional shift. We also show that the strength of the conditional shift in the data-generating process determines the performance gap between our model and other group disentanglement methods. Our evaluation is run on the downstream task of extracting student aptitudes from a dataset of test scorse grouped by school, a problem on which group-instance models have not been applied before.

The main limitation of our work is that we perform evaluation on a synthetic dataset of student scores rather than real data. Although this is a justifiable choice with respect to evaluation (it gives us access to the ground-truth values of the latent variables), future work should focus on evaluating on real-world datasets with conditional shift, such as user-item ratings (Koren et al., 2009).

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
