# OpenReview forum: "Group-Disentangling Conditional Shift"
_ICLR.cc/2023/Conference — Submitted to ICLR 2023_

### Official Review · Reviewer_KtA6 · 2022-10-24

**Confidence:** 5
**Correctness:** 4
**Technical Novelty And Significance:** 2
**Empirical Novelty And Significance:** 2
**Recommendation:** 3

**Clarity, Quality, Novelty And Reproducibility:**

The paper is well written with clear demonstration of the problem and solution. The work is partially original but there are many existing models using the similar ideas. The code is not available thus cannot demonstrate its reproducibility.

**Strength And Weaknesses:**


The major strength of this paper can be summarized in the following aspects:

1. This paper correctly points out the weakness of existing methods in group disentanglement when the conditional shift exists. The idea that the group variables are confounders when inferring the individual variables is important to know.

2. Learning disentangled representation to handle the conditional shift is relatively new since as the paper stated, most methods focus on learning invariant representation under the shift.

3. The synthesis examples are easy to follow and it demonstrates the impact of conditional shift over existing group entanglement methods

The major weakness of this paper can be summarized in the following aspects:

1. The idea of conditioning/controlling on group variables when inferring the individual variables is not novel. It follows naturally by the definition of conditional shift, i.e. the group conditional distribution of instances are different.  As the paper points that it is widely used in semi-supervised learning. The innovation point is its use in unsupervised learning and generative models. However, it is not sufficient to meet the bar for ICLR.


2. One of main concerns for the conditioning method is when it deals with high dimensional problems. In high dimensional space, conditioning would restrict the set of samples that are available to the model in each group. Note that this method essentially learns a set of group specific models. In high dimensional setting, the generative model needs a lot more samples generated before a robust inference result is obtained. This is partially why the conditional independence assumption was used, since it would save a lot of time in data generation.


3. That being said, the experiments are too simple. It is a low dimensional example, while the high dimensional data set such as image dataset are mentioned but not tried. It is important to demonstrate the strength and weakness of this method in high dimensional setting, esp the time for inference and the robustness of the inference.


4. No code provided. Although this is not hard to implement, it is better to have some code to prove the reproducibility.


**Summary Of The Paper:**

This paper proposed a novel group entanglement method under the concern of conditional shift in dataset. In the paper, the author argues that under the conditional shift, the group representation and instance representation cannot be inferred independently since the instance distribution is confounded by the group identity. The proposal is to control the group variable while learning the individual representation. This paper claims to be the first in unsupervised group disentanglement to condition on group variables while learning the individual variables.


**Summary Of The Review:**

In sum, this paper provides an interesting perspective on the group entanglement under conditional shift. The solution is intuitive and easy to follow. However, the idea is not very novel since it has been explored in many tasks before. It also fails to demonstrate its strength and weakness under more realistic and high dimensional dataset.

---

### Official Review · Reviewer_iRs6 · 2022-10-24

**Confidence:** 4
**Correctness:** 2
**Technical Novelty And Significance:** 2
**Empirical Novelty And Significance:** 2
**Recommendation:** 3

**Clarity, Quality, Novelty And Reproducibility:**

**Clarity and Writing issues**
- Overall the writing in the introduction is not easy to follow. There is no clear flow between problems tackled in the paper, issues with the existing works, and contributions of the paper.
- The writing in contribution bullets is a bit hard to follow. The first bullet " We approach the task of learning fair representations of students from different schools/socio-economic backgrounds." appears to be a bit disconnected from the previous sentence.
- "conditional shift directly causes our model’s improvement in performance over existing methods", is a bit misleading. This statement can not be true in general without the additional assumptions on what is not shifting. Moreover, the sentence structure is also unnecessarily complex.


**Reproducibility concern**
- Code or detailed experiment setting is not provided in the paper.
- Moreover, no hyperparameter details are shared

**Strength And Weaknesses:**

**Strength**
- Paper tackles an important and relevant problem

**Weaknesses**
- Results are present only on the toy dataset in the paper. This is the biggest weakness of the paper. Moreover, since the data-generating process is also proposed in the paper, it is unclear if the dataset is specially designed that can show the failure modes of other methods and if those failure modes are present in other real-world datasets.
- Since all the experiments are on toy datasets, the claims made in the abstract and introduction are overstated. For example, "Our model has the novel ability to disentangle ambiguous observations". There is no concrete evidence in the paper when this will hold and how general of a statement this is?
- Tackling the problem of conditional shift is very general and ill-posed. It is unclear from the writing how the paper deals with inherent underspecification.
- Method description in Section 4 is a bit skim. Equations 8-10 appear to be a bit out of the place and it is unclear how the text above these equations follows.
- Only the toy dataset is considered in the paper. Any description evaluation criterion is missing. It is hard to understand the tasks considered in Figures 3 and 4.
- Reproducibility statement is not present and No code is provided as well. Authors can use the 9th page in the main paper and additional appendices to provide those details.

**Summary Of The Paper:**

The paper tackles the problem of group disentanglement in presence of conditional shift. This work proposes a new group disentanglement method called the Context-Aware Variational Autoencoder. Experiments on toy datasets show that the proposed method can significantly improve over existing methods.

**Summary Of The Review:**

Overall, the writing of the paper is very unclear and the proposed method is only evaluated on toy datasets. It is also unclear how the inherent underspecification of conditional shift is handled in the paper.

---

### Official Review · Reviewer_G1Wn · 2022-10-25

**Confidence:** 3
**Correctness:** 2
**Technical Novelty And Significance:** 3
**Empirical Novelty And Significance:** 3
**Recommendation:** 3

**Clarity, Quality, Novelty And Reproducibility:**

The paper is written well. The idea is simple and exists in prior work, but the novelty seems to be in using the group-representation-conditioning to better learn instance representations.

**Strength And Weaknesses:**

Strengths:

1. Simple modification to the posterior affords good advantanges.
2. Promising performance on synthetic data.


Weaknesses:

My main concern with the paper is acknowledged by the authors but nonetheless remains important: "The main limitation of our work is that we perform evaluation on a synthetic dataset of student scores rather than real data."

I do not think such an evaluation can be avoided. Reconstruction error is the one metric that I can trust and that to me only sounds like a part of the story in the paper. Translation additionally seems to be important but I do not see it evaluated on real data.

Important questions include:

1. What is the point out the translation metric if it cannot be evaluated on real data? The authors say "Our model preserves the relative positions of the scores" in figure 2. It this something we desire naturally or something that comes out of an assumption?
2. How can we guarantee relative positions of the features when translating without restrictions on q?
3. What the desiderata for disentanglement here without stating the method? How should one evaluate them?
4. The definition of conditional shift seems to say "changing the group changes the instance representation for the same features.". This is a natural consequence of conditioning on the collider as in figure 1 first figure (assuming a causal graph). Why call it conditional shift when it's a consequence of the assumed data generating process?


**Summary Of The Paper:**

The paper makes a simple modification to the parameterized posterior distribution to allow for learning group-representations and the within-group instance representation when they are dependent conditional on the observed features: condition the instance representations on both the features and the group representation. As the paper puts it, this can handle conditional shift where changing the group changes the instance representation for the same features.

**Summary Of The Review:**

The paper is written well, but it remains to be seen whether the proposed method is useful for any real datasets.

---

### Official Review · Reviewer_sLYB · 2022-10-25

**Confidence:** 2
**Correctness:** 2
**Technical Novelty And Significance:** 3
**Empirical Novelty And Significance:** 2
**Recommendation:** 5

**Clarity, Quality, Novelty And Reproducibility:**

The paper is clear, but lacking of intuition. For example,
1. why we need to add $u_n$ into distribution $Q$? Any intuition for doing that?
2. What is the proof detail for eqn 8-10? Some equation is wield. Eqn 3-4 are also the same equation.
3. It is unclear how to implement the proposed ELBO loss in real world?  Which reparameter trick are you using?
4. Why the method was only tested in synthetic data? How about high dimensional real world images? For example, the GVAE tested in image data. It is conventional to show in some real world high dimensional data.

**Strength And Weaknesses:**

Strength:
- the work touches a fundamental problem.

Weakness:
- Only synthetic experiments are conducted.
- The VAE only tested with MLP.
- The data generated is in low dimensional and not very persuasive.

**Summary Of The Paper:**

The paper proposes a context aware variational auto encoder which modified the structure of previous C-VAE. The evaluation is on synthetic data only.

**Summary Of The Review:**

I think the paper is lacking of intuition and details at this stage. In addition, the experiment is insufficient (only synthetic data used).

---

### Author Response · Authors · 2022-11-18
**Paper Update**

We thank the reviewers for their careful reading of the paper and for their insightful and constructive comments. We agree with the majority of their comments and we recognise the need for substantially updating this paper.

- In our updated paper, we focused on fixing the main limitation identified by the reviewers: the lack evaluation on high-dimensional datasets. To address this, we tested our model on the 3DIdent ( [https://arxiv.org/abs/2102.08850](https://arxiv.org/abs/2102.08850) ) dataset, a popular dataset for evaluating disentanglement that also exhibits conditional shift. We observe our model producing a significant improvement in disentanglement over prior work. The new results can be seen in Section 6.

We also agree with the many useful secondary comments that we did not manage to address in time for the rebuttal deadline. Those will be our main focus for the next version of the paper:

- The need for more intuitive explanation of why conditioning on the group variable is necessary.
- Adding a code repository as supplementary material that would allow readers to reproduce the experiments.
- A discussion of the assumptions of conditional shift and a formalisation of the space of problems we are considering.

In the new version of the paper, we also make a few small corrections pointed out by the reviewers.

- Added optimiser hyperparameters in the "Model Setup" section.
- Used a single number to label multi-line equations.

---

### Decision · Program_Chairs · 2023-01-20

**Decision:**

Reject

**Justification For Why Not Higher Score:**

This paper addressed an important problem in ML and demonstrated promising results on synthetic data. However, the reviewers shared concerns on the motivation of the proposed idea, missing discussion on the assumptions of conditional shift, and missing implementation details for reproduction. Although the authors addressed the reviewers' concern on synthetic data in their rebuttal, the reviewers were not convinced after the rebuttal.
I think this paper is not ready for publication at the current stage due to missing details and the weaknesses listed in the summary.

**Justification For Why Not Lower Score:**

N/A

**Metareview: Summary, Strengths And Weaknesses:**

This paper proposes a group disentanglement method called the Context-Aware Variational Autoencoder(CxVAE). The authors argued that under the conditional shift, the group variable and instance variable cannot be inferred independently since the instance distribution is confounded by the group identity. They propose to control group variable while learning individual representations.  Experiments on synthetic datasets show that the proposed method can improve over existing methods.

Strengthes:
- The paper addressed a fundamental problem in ML.
- The results on synthetic data are promising.

Weaknesses:
- It's not well-motivated. The authors did not provide an intuitive explanation of why conditioning on the group variable is necessary.
- It does not discuss the assumptions of conditional shift and a formalization of the space of problems.
- The experimental results are not reproducible due to missing implementation details.
- It was tested only on synthetic data. The authors added experiments on 3DIdent in their revised version.